# Lactic Acid Bacteria and Bioactive Amines Identified during *Manipueira* Fermentation for *Tucupi* Production

**DOI:** 10.3390/microorganisms10050840

**Published:** 2022-04-19

**Authors:** Brenda de Nazaré do Carmo Brito, Renan Campos Chisté, Alessandra Santos Lopes, Maria Beatriz Abreu Gloria, Gilson Celso Albuquerque Chagas Junior, Rosinelson da Silva Pena

**Affiliations:** 1Graduate Program in Food Science and Technology (PPGCTA), Institute of Technology (ITEC), Federal University of Pará (UFPA), Belém 66075-110, PA, Brazil; brendabrito07@gmail.com (B.d.N.d.C.B.); rcchiste@ufpa.br (R.C.C.); alessalopes@ufpa.br (A.S.L.); chagasjunior.gca@gmail.com (G.C.A.C.J.); 2Faculty of Food Engineering (FEA), Institute of Technology (ITEC), Federal University of Pará (UFPA), Belém 66075-110, PA, Brazil; 3Laboratory of Quality Control (LQC), Faculty of Pharmacy, Federal University of Minas Gerais (UFMG), Belo Horizonte 31270-901, MG, Brazil; mbeatriz.gloria@gmail.com

**Keywords:** *Lactobacillus*, *Manihot esculenta*, cassava product, Amazonian fermented product, fermented food

## Abstract

There is scarce information regarding lactic acid bacteria (LAB) and the production of biogenic amines during *manipueira* fermentation for *tucupi*. Thus, the objective of this study was to isolate and identify LAB, and to determine their impact on bioactive amine formation. Spontaneous fermentation of *manipueira* was carried out at laboratory scale and selected LAB colonies were isolated and identified by sequencing techniques and comparison with sequences from a virtual database. Only two LAB species of the genus *Lactobacillus* were identified during fermentation: *Lactobacillus fermentum* and *Lactobacillus plantarum*. *L. fermentum* was the predominant, whereas *L. plantarum* was only detected in *manipueira* prior to fermentation. Spermidine and putrescine were detected throughout fermentation, whereas histamine was produced at the final stage. There was positive correlation between LAB counts and putrescine and histamine levels, suggesting that the identified LAB are responsible for the synthesis of these amines during *manipueira* fermentation. Genetic assays are needed to verify whether the LAB identified have the genes responsible for decarboxylation of amino acids.

## 1. Introduction

Cassava roots (*Manihot esculenta* Crantz) are traditionally used in Northern Brazil to produce several derived food products, with cassava flour one of the main target products. During cassava flour production, the crushed cassava roots are pressed to remove a liquid fraction called *manipueira* (also known as wastewater), resulting from the extraction of cassava starch [1,2]. *Manipueira* is separated from the cassava and subjected to spontaneous fermentation for up to 24 h, followed by cooking, to produce a broth named *tucupi* [3,4]. *Tucupi* is a traditional ingredient from Amazonia (mainly Northern Brazil) that is widely used in the regional cuisine and with great potential for use in the food industry [5].

In a recent study from our group [6], *manipueira* fermentation was characterized by two distinct stages: the first (up to 12 h fermentation) was characterized by higher pH values, and higher contents of total and reducing sugars, and total starch, whereas the second stage (16 to 24 h) was characterized by the production and buildup of biogenic amines (histamine-HIM and tyramine-TYM). The resulting *tucupi* had a high acidity and soluble solids. The polyamine spermidine (SPD) and its obligate precursor putrescine (PUT) were present in the original *manipueira* and remained at similar levels throughout fermentation. In the second stage of fermentation, from 16 h on, there was production and buildup of the biogenic amines, TYM and HIM. Boiling of the fermented *manipueira* after fermentation did not significantly affect amine levels. It is likely that the production of biogenic amines results from amino acid decarboxylase activity from the fermentation microbiota, with the amines surviving heat treatment.

There is scarce information in the literature regarding the microbiota involved during *manipueira* fermentation to produce *tucupi.* Studies on the microbiota of wastewater from cassava described the occurrence of lactic acid bacteria (LAB) and yeast species [7]. Among LAB, the genus *Lactobacillus* was reported to be predominant [1,2]. LAB play an important role in fermentation processes as they intensify fermentation and are responsible for the acidification of the medium, which contributes to the desirable characteristics of the fermented products. In addition, the acidification of the medium limits the growth and survival of pathogenic microorganisms contributing to safety [7,8]. However, LAB can have genes responsible for the synthesis of biogenic amines, and the presence of some of them (e.g., HIM and TYM) at high concentrations is usually not desirable in food products, as they can cause adverse effects to human health [9].

The scarcity of data in the literature regarding the microorganisms involved during the fermentation of *manipueira* to produce *tucupi*, and their role in biogenic amine formation and accumulation, motivated this research. In this context, the objective of this study was to isolate and identify LAB and to quantify bioactive amines during *manipueira* fermentation. These findings can provide information for the mitigation of biogenic amine accumulation in *tucupi* and other fermented cassava products.

### 1.1. Raw Material

Yellow pulp cassava roots (30 kg) were used. This type of cassava is the most widely used in the production of *tucupi* in Northern Brazil. The samples were acquired in the *Ver-o-Peso* market (Belém, Pará, Brazil) (01°27′08″ S, 48°30′13″ W), and the collections were carried out in three different periods (*n* = 3): the months of October, November and December (10 kg per batch).

### 1.2. Manipueira’s Fermentation

To obtain *manipueira*, the cassava roots were washed with potable water, peeled with stainless steel knives, and washed again with potable water to eliminate undesirable solid residues. The peeled roots were then grated and the mass obtained (8 kg per batch) was manually pressed to extract *manipueira* (4 L per batch). These steps (washing, peeling, crushing and pressing) were carried out at the sample acquisition place, using the same equipment and processing conditions as the traditional *tucupi* producers in the market. The *manipueira* was transported to the laboratory at Federal University of Pará (UFPA) in sterile plastic containers with a capacity of 5 L, at room temperature (≈25 °C). Sample transportation to the laboratory took ca. 30 min.

Each *manipueira* batch (4 L) was transferred to a stainless container, previously sanitized with sodium hypochlorite solution (100 mg/L for 30 min) and rinsed with potable water to eliminate any sodium hypochlorite residue. Then, *manipueira* was subjected to spontaneous fermentation at 30 °C (average room temperature in the region) for 24 h, in an oven (Q-316 M5, Quimis, Diadema, SP, Brazil). To monitor the process, aliquots were removed at 0, 12 and 24 hours of fermentation. Microbiological analyses were carried out immediately after aliquot collection and the samples used for the determination of bioactive amines were placed in 50 mL plastic containers and stored under freezing (−18 °C) until analysis. These steps were repeated with samples from each batch (*n* = 3).

### 1.3. Isolation of Lactic Acid Bacteria (LAB)

For the isolation of LAB, 25 mL of *manipueira* (collected at the three fermentation sampling times) were homogenized in 225 mL 0.1% saline peptone water (Kasvi, São José dos Pinhais, PR, Brazil) and serially diluted to a 10^−8^ dilution. For LAB enumeration, the *pour plate* inoculation technique was used in sterile petri dishes, containing De Man, Rogosa & Sharpe (MRS) agar (pH 6.2, Kasvi), with 0.2% nystatin (Pratti-Donaduzzi, Toledo, PR, Brazil) for fungal growth inhibition. The plates were inverted for incubation in a bacteriological incubator (DeLeo, Porto Alegre, RS, Brazil) at 37 °C for 72 h [10].

From each plate containing 25 to 250 colonies, 35% of the colonies were removed [11] and submitted to the gram stain and catalase tests, using 2% hydrogen peroxide (Neon, Brazil). Negative catalase and gram-positive colonies with cocci or bacilli morphology were isolated and purified twice by the depletion technique in petri dishes containing MRS agar (Kasvi, pH 6.2). An overlay of the same medium was applied to ensure an anaerobic medium, and the sets were incubated at 37 °C for 72 h. The counts were expressed in log Colony Forming Units per milliliters of sample (log CFU/mL).

### 1.4. Molecular Identification

#### 1.4.1. DNA Extraction

The genomic DNA of the isolated and purified colonies was extracted according to the phenol/chloroform/isoamyl alcohol protocol (25:24:1, *v*/*v*/*v*, Sigma-Aldrich Co., St. Louis, MI, USA) by Sambrook and Russell [12] with subsequent suspension in 100 µL of Tris-EDTA buffer (TE, pH 6.0) and freezing at −18 °C until further PCR steps were performed.

#### 1.4.2. Polymerase Chain Reaction (PCR)

The bacterial region of the 16S gene was amplified by the PCR technique, in which 2 μL DNA, previously thawed to room temperature (≈25 °C) was added to 23 μL solution containing 16.4 μL sterilized purified water (Millipore Corp., Milford, MA, USA), 2.5 μL 10× buffer solution (Invitrogen, Carlsbad, CA, USA), 1 μL dNTP mix (Qiagen, Hilden, Germany), 2 μL MgCl_2_ (50 mM, Invitrogen), 0.1 μL Taq DNA polymerase (5 U/μL, Invitrogen), 0.5 μL of primer 616F (Invitrogen) (5′-TTAAAAVGYTCGTAGTYG-3′) and 0.5 μL of primer 907R (Invitrogen) (5′-CCGTCAATTCMTTTGAGTTT-3′), in a concentration of 10 pmol each [13,14].

The microtubes were subjected to PCR in an automatic thermocycler (mod. K960, LabTrace, China) programmed for initial denaturation reactions at 94 °C/4 min, 30 denaturation cycles (94 °C/15 s), annealing (55 °C/2 min), extension (72 °C/3 min), final extension (72 °C/10 min) and cooling (4 °C) [14].

Afterwards, PCR products were purified with the Axygen PCR-clean up kit (mod. AP-PCR-250, Glendale, AZ, USA), following manufacturer’s instructions. Molecular identification was performed by bidirectional-sequenced reactions, with analysis performed in an ABI-Prism 3700 Genetic Analyzer sequencer (Applied Biosystems, Foster City, CA, USA).

The quality of the sequences was evaluated using the FinchTv 1.4.0 software (Geospiza Inc., Seattle, WA, USA) and the GenBank virtual database (http://blast.ncbi.nlm.nih.gov, accessed on 1 March 2019) was used to compare the degree of identity.

### 1.5. Bioactive Amine Determination

For the determination of free bioactive amines, 5 mL of *manipueira* was used. After homogenization in a shaker, the sample was submitted to centrifugation (Jouan Thermo MR23i, Saint-Herblain, France) at 11,180× *g* at 4 °C for 10 min. The supernatant was filtered through qualitative filter paper and then on a 0.45 μm membrane, prior to high performance liquid chromatography (HPLC) injection [15].

For the determination of bioactive amines, ion-pair HPLC and fluorescence detection were used. A model LC-10CE chromatograph with a high-pressure mixing chamber, automatic piston washer assembly and model SIL-10AD VP automatic injector (Shimadzu, Kyoto, Japan) was used. The compounds were separated using a Novapak C18 column (3.9 × 300 mm, 4 µm, 60 Å, Waters Co., Milford, MA, USA) and an elution gradient of 0.2 M sodium acetate and 15 mM sodium octanesulfonate, pH adjusted to 4.9, with glacial acetic acid (mobile phase A) and acetonitrile (mobile phase B) [15].

The identification of bioactive amines was performed by comparing retention times and co-elution with standards of spermidine, putrescine, agmatine, cadaverine, serotonin, histamine, tyramine, tryptamine and phenylethylamine. Quantification was performed by fluorimetry (340 and 445 nm excitation and emission, respectively), after post-column derivatization with o-phthalaldehyde, using external analytical curves for each compound (r^2^ ≥ 0.99). The content of bioactive amines (*n* = 3) was expressed in mg/L of sample.

### 1.6. Statistical Analysis

The results were submitted to analysis of variance (ANOVA) at 5% significance, and the means were compared by the Tukey’s test (*p* ≤ 0.05). Pearson’s correlation was carried out to investigate existence of correlation between the LAB count and amine levels. Statistica 7.0 software (StatSoft Inc., Tulsa, OK, USA) was used.

## 2. Results

Nineteen LAB were isolated during the spontaneous fermentation of *manipueira*, at times 0 h (*n* = 10), 12 h (*n* = 4) and 24 h (*n* = 5) (Table 1). Regarding LAB, the counts ranged from ~7 and ~9 log CFU/mL during fermentation until the end of the process. The LAB count in the *manipueira* (0 h) was 7.2 log CFU/mL; at 12 h of processing the bacterial population increased to 8.2 log CFU/mL; and then remained the same (*p* > 0.05) until the end of fermentation–24 h (8.8 log CFU/mL). These results confirm the presence of LAB in *manipueira*, before and during the fermentation process.

Two species of LAB, belonging to the genus *Lactobacillus,* were identified through sequencing techniques and comparison of sequences in GenBank virtual databases, at a level greater than 95% of similarity: *L. fermentum* and *L. plantarum*. The distribution of these species during fermentation showed that *L. plantarum* was present only at the beginning of fermentation and represented 10% of the LAB. The remaining 90% of the LAB were represented by *L. fermentum* (Table 1). This bacterium was present at the beginning (0 h), in the middle (12 h) and at the end of fermentation (24 h), and was the only species of LAB detected after 12 h of fermentation.

During spontaneous fermentation of *manipueira*, three bioactive amines were identified: one polyamine (spermidine–SPD) and two biogenic amines (putrescine–PUT and histamine–HIM) (Figure 1). SPD and PUT were detected in all the stages of *manipueira* fermentation, with average concentrations of 2.88 mg/L and 2.72 mg/L, respectively. HIM was only detected at the end of the fermentation process (24 h) at levels of 1.33 mg/L.

There was significant positive correlation (Pearson’s correlation) between amines and LAB counts, with correlation coefficients of 0.98 for PUT and 0.78 for HIM. Spermidine levels correlated negatively with LAB counts (r = −0.83), which may be related to nitrogen source consumption by the microbiota present in the process.

## 3. Discussion

Fermentation of *manipueira* to produce *tucupi* is a spontaneous process, which the microbiota has not yet been thoroughly elucidated. Thus, this is the first study that identified the LAB involved in the fermentation process of *manipueira* for the production of *tucupi*. Emphasis was given to LAB since they were reported to be the predominant class of microorganisms in the fermentation of cassava-based foods [16] and wastewaters [1,2,7]. Additionally, LAB have the ability to produce biogenic amines [17].

Lactic fermentation is the predominant type of fermentation reported for cassava products, but there are reports of the presence of other types of microorganisms at the beginning of the process [18,19,20]. As fermentation progresses, there is a succession of other microorganisms such as LAB, predominantly the genus *Lactobacillus*. This behavior can be attributed to the sensitivity of other microorganisms present at the beginning of fermentation, and to the acidic conditions of the medium, which takes place with the advancement of fermentation [21].

The findings regarding the types of bioactive amines present during *tucupi* production are in agreement with Brito et al. [6]. However, they also detected tyramine, which was not detected in this study. Since the studies were similar, involving the same variety of cassava and processing conditions, though not the same time period (April, May and June), differences in results could be associated with environmental sources [22].

Following the same trend observed by Brito et al. [6], no significant changes in the contents of SPD and PUT (*p* > 0.05) were observed throughout fermentation. The presence of these amines was expected since they are inherent to all living organisms. SPD is involved in cell growth, renewal and metabolism; and PUT is an obligatory precursor to the formation of polyamines [23,24]. However, the levels of these amines were lower than those reported by Brito et al. [6], who observed mean concentrations of 4.43 mg/L for SPD and 3.78 mg/L for PUT during *manipueira* fermentation. The observed differences can be attributed to the fact that the polyamines and their precursor can be synthesized and utilized by the microorganisms, as the polyamines are important for growth and development [25].

In a similar way, the occurrence and levels of the biogenic amines HIM and TYM (1.33 mg/L and nd—not detected, respectively) were also lower in this study compared to Brito et al. [6]. LAB are capable of producing TYM and HIM, as a protection response against medium acidification (pH decrease) that occurs during the fermentation process [25].

The negative correlation between LAB counts and SPD levels suggests the use of this polyamine as a growth factor; however, the positive correlation between LAB, and PUT and HIM, suggests the potential formation of these amines by LAB in *tucupi* production.

The species *L. fermentum* and *L. plantarum*, identified during the fermentation of *manipueira,* have the capacity to decarboxylate amino acids, being able to produce the amines HIM, TYM and PUT [25,26]. The distribution of LAB (Table 1) and the profile of bioactive amines (Figure 1) observed, provided strong indications that *L. fermentum* and/or *L. plantarum* were responsible for the production of PUT, and *L. fermentum* was responsible for the production of HIM during the production of *tucupi*. Landete et al. [27] observed 100% correlation between the presence of histidine decarboxylase, tyrosine decarboxylase and ornithine decarboxylase genes, and the production of HIM, TYM and PUT, respectively. Thus, further study is needed to confirm whether the genes that encode the decarboxylase enzymes, which are responsible for the synthesis of biogenic amines, are present in the *L. fermentum* and *L. plantarum* identified during *manipueira* fermentation.

The results of the study indicated that the two LAB species identified might be responsible for the production of putrescine and histamine, during the fermentation process. However, genetic studies are needed to prove the ability of these bacteria to encode decarboxylases, involved in the synthesis of the identified biogenic amines. Studies investigating yeast during the spontaneous fermentation of *manipueira* during *tucupi* production are also of paramount importance.

The formation and buildup of HIM and TYM should be prevented, since these amines are resistant to any heat treatment [6]. In addition, at high concentrations, these amines can cause adverse effect to human health. The levels of these amines in *tucupi* are below no adverse effect levels (NOAEL) for normal individuals. However, individuals who are sensitive to HIM should avoid any HIM in meals. With respect to TYM, individuals under monoaminoxidase inhibitor (MAOI) drugs should also avoid TYM in meals to prevent migraines and hypertensive crisis [28,29].

## 4. Conclusions

This is the first study in which lactic acid bacteria (LAB) active in the fermentation of *manipueira* for *tucupi* production were identified. *Lactobacillus plantarum* and *L. fermentum* were the two LAB species identified during the fermentation process, but *L. plantarum* was only identified in *manipueira* prior to fermentation. The counts of the LAB increased during fermentation. Spermidine (polyamine) and putrescine (polyamine precursor) were detected throughout the fermentation process, at similar levels. The biogenic amine histamine was detected when LAB counts reached 8.8 log CFU/mL, at the end of the fermentation. There was significant positive correlation between LAB counts and putrescine and histamine levels, and negative correlation with spermidine, suggesting the role of LAB in amine formation and accumulation in *tucupi* production.

Regarding the amounts of bioactive amines determined in this study, *tucupi* proved to be a safe food for consumption, with values below the limits established for PUT, HIM and TIM. However, the manufacturing techniques (still rudimentary) need greater attention from investors to avoid the risk of increased microbial contamination during the process, preventing the production of undesirable compounds in the product.

## Figures and Tables

**Figure 1 microorganisms-10-00840-f001:**
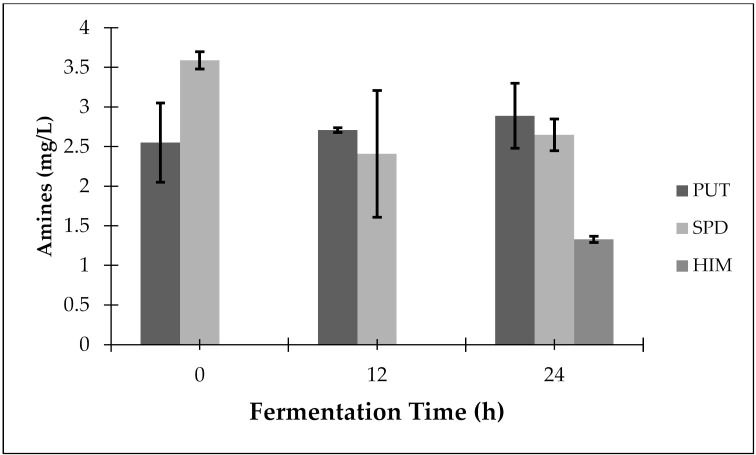
Levels of bioactive amines during the spontaneous fermentation of manipueira for *tucupi* production. HIM: histamine; PUT: putrescine; SPD: spermidine.

**Table 1 microorganisms-10-00840-t001:** Molecular identification of lactic acid bacteria isolated during the spontaneous fermentation of *manipueira* for *tucupi* production.

Fermentation Time	LAB Count ^1^ (log CFU/mL)	Lactobacilus Species	GenBank Access	Similarity (%) ^2^
0 h	7.2 ± 0.1 ^b^			
		*L. fermentum*	MK640639.1	
2		*L. fermentum*	MK616469.1	100
3		*L. fermentum*	MK640639.1	99
4		*L. fermentum*	MK640639.1	100
5		*L. fermentum*	MK640639.1	93
6		*L. fermentum*	MH817767.1	98
7		*L. fermentum*	MK640639.1	99
8		*L. fermentum*	MK640639.1	99
9		*L. plantarum*	MK616469.1	98
10		*L. fermentum*	MK639007.1	99
12 h	8.2 ± 0.1 ^a^			
1		*L. fermentum*	MK640639.1	100
2		*L. fermentum*	MK640639.1	100
3		*L. fermentum*	MK639007.1	99
4		*L. fermentum*	MK640639.1	99
24 h	8.8 ± 0.1 ^a^			
1		*L. fermentum*	CP035055.1	96
2		*L. fermentum*	MF108112.1	96
3		*L. fermentum*	MH175491.1	94
4		*L. fermentum*	MK640639.1	99
5		*L. fermentum*	MK640639.1	99

^1^ mean ± standard deviation. ^2^ Percentage of similarity with lactic acid bacteria (LAB) available in GenBank database. Means with the same superscript letters in the same column are not statistically different (*p* > 0.05) (Tukey’s test).

## Data Availability

Not applicable.

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
