# Peer review of "Lactic Acid Bacteria and Bioactive Amines Identified during Manipueira Fermentation for Tucupi Production"

_microorganisms, 2022, doi:10.3390/microorganisms10050840_

Round 1

Reviewer 1 Report

The authors identified Lactobacillus fermentum and Lactobacillus plantarum based on molecular techniques during the fermentation of manipueira from Cassava roots during the production of the traditional food ingredient tucupi. Spermidine and putrescine were detected during this fermentation. The paper should be revised to improve the English language usage and grammar particularly in the Abstract and Introduction. Please address why they conclude that this is the first study to identify lactic acid bacteria active in manipueira fermentation when references 8 and 9 reported Lactobacillus in manipueira Cassava waste fermentation. Also, how does knowledge of these two LAB species improve the production of tucupi and if the levels of spermidine and putrescine represent a health risk or negative organoleptic properties for the consumer. It would also be a good idea to report changes in species diversity during manipueira fermentation. Finally, is Cassava starch the primary substrate during manipueira fermentation by Lactobacillus fermentum and Lactobacillus plantarum?

Author Response

Response to Reviewers’ comments

  1. Ref. No.: microoganisms-1618451

Title: “Lactic acid bacteria and bioactive amines identified during manipueira fermentation aiming tucupi production”

Dear Editor,

Attached, please find the revised manuscript with the requested clarifications and changes. We thank the editor and reviewers for taking the time and for the careful reading and evaluation of this manuscript. We appreciated the suggestions and we are certain that the manuscript has been enhanced with the changes.

Our responses to the recommendations are listed follows.

Best regards.

Reviewer 1

  1. Comment: The authors identified Lactobacillus fermentum and Lactobacillus plantarum based on molecular techniques during the fermentation of manipueira from Cassava roots during the production of the traditional food ingredient tucupi. Spermidine and putrescine were detected during this fermentation. The paper should be revised to improve the English language usage and grammar particularly in the Abstract and Introduction.

Our reply: Thanks for evaluate the manuscript of this research. The English language was carefully improved by a native English speaker.

  1. Comment: Please address why they conclude that this is the first study to identify lactic acid bacteria active in manipueira fermentation when references 8 and 9 reported Lactobacillus in manipueira Cassava waste fermentation. Also, how does knowledge of these two LAB species improve the production of tucupi and if the levels of spermidine and putrescine represent a health risk or negative organoleptic properties for the consumer. It would also be a good idea to report changes in species diversity during manipueira fermentation.

Our reply: This is a good point to be explained. This is the first study about the lactic acid bacteria in fermentation of a waste cassava for production of a food very appreciated in Amazon Region: tucupi. The mentioned references are not based on manipueira or tucupi, but cassava water waste (residue or another type of food – fofo from Africa by Abriba et al.).

Regarding the safety aspects associated with biogenic amines, the concerns are associated with histamine (and tyramine), which, at high concentrations, can cause adverse effects to human health. Histamine levels would be a concern to histamine sensitive individuals. This information was added to the manuscript (page 1, lines 62-66 and page 7, lines 1-7).

Spermidine and putrescine are naturally present in any living matter. As mentioned in the manuscript, spermidine is a growth factor, essential for several physiological functions in plants, animals and microorganisms (page 6, lines 21-24). They are synthesized as needed by the microorganism. Putrescine is an obligate intermediate of spermidine and may accumulate under stressful conditions. During manipueira fermentation (this study and also Brito et al.) the levels were kept unchanged, suggestion the use and production by the microorganisms to keep ideal growth conditions.

Ref.:

- Abriba. C.; Henshaw, E.E.; Lenox, J.; Eja, M.; Agbor, B.E. Microbial succession and odour reduction during the controlled fermentation of cassava tubers for the production of ‘fofo’, a staple food consumed popularly in Nigeria. J. Microbiol. Biotecnol. 2012, 2, 500-506.

- Brito, B. de N. do C.; Chisté, R.C.; Lopes, A.S.; Glória, M.B.A.; Pena, R. da S. Influence of spontaneous fermentation of manipueira on bioactive amine and carotenoid profiles during tucupi production. Food Res. Int. 2019, 120, 209–216, doi.org/10.1016/j.foodres.2019.02.040.

- EFSA Panel on Biological Hazards (BIOHAZ). Scientific Opinion on risk based control of biogenic amine formation in fermented foods. EFSA J. 2011, 9, 1-93. doi.org/10.2903/j.efsa.2011.2393.

  1. Comment: Finally, is Cassava starch the primary substrate during manipueira fermentation by Lactobacillus fermentum and Lactobacillus plantarum?

Our reply: This is a good question. During manipueira fermentation, the starch naturally found in cassava roots is decanted, which suggests that it does not serve as a substrate for LABs.

Reviewer 2 Report

Well written, concise and easy to understand work.

As for the results presented by the authors, the standard deviation values ​​should be introduced, both for the LAB and for the amines. As well as the results of the statistical analysis described in the material and methods section. In spontaneous fermentations, the growth of microorganisms can be quite variable, so it is important to correlate the number of microorganisms present with the amines produced and, at the same time, verify that the fermentations between the different batches were similar. In the analysis of amines, it is also important to present the percentage of recovery of the amines in the sample, as the extraction method used may not be able to extract all the amines present.

Author Response

Response to Reviewers’ comments

  1. Ref. No.: microoganisms-1618451

Title: “Lactic acid bacteria and bioactive amines identified during manipueira fermentation aiming tucupi production”

Dear Editor,

Attached, please find the revised manuscript with the requested clarifications and changes. We thank the editor and reviewers for taking the time and for the careful reading and evaluation of this manuscript. We appreciated the suggestions and we are certain that the manuscript has been enhanced with the changes.

Our responses to the recommendations are listed follows.

Best regards.

Reviewer 2

  1. Comment: Well written, concise and easy to understand work.

Our reply: We appreciate the positive feedback from the reviewer. Thank you very much for your attention to evaluate this manuscript.

  1. Comment: As for the results presented by the authors, the standard deviation values should be introduced, both for the LAB and for the amines. As well as the results of the statistical analysis described in the material and methods section.

Our reply: The means and standard deviation of LAB counts were added to Table 1. The same values were added for amines in Figure 1.

  1. Comment: In spontaneous fermentations, the growth of microorganisms can be quite variable, so it is important to correlate the number of microorganisms present with the amines produced and, at the same time, verify that the fermentations between the different batches were similar.

Our reply: Thanks for the observation. Our study studied the fermentation of manipueira on a laboratory scale, simulating (in a way) how it is carried out in traditional commercial places. The LAB counts were similar among the three batches, as can be seem in Table 1 (CV < 1.3%) and there was significant correlation of LAB counts with the amines content (Pearson correlation). Thanks for reminding us about correlation studies which provided additional relevant information in our study.

  1. Comment: In the analysis of amines, it is also important to present the percentage of recovery of the amines in the sample, as the extraction method used may not be able to extract all the amines present.

Our reply: Thanks for the question. We realized that there was a mistake in the method reported. TCA has been widely used for the extraction of amines from solid foods (fish, cheese, sausage, cocoa, chocolate, etc.). However, in liquid samples like tucupi (wine, beer, and other beverages), the sample is only homogenized and centrifuged, obtaining full recovery of free amines.

The method for the analysis of amines was corrected and the word ‘free’ was added to the amines.

Reviewer 3 Report

The paper addresses isolation of bacteria and amines from a traditional procedure on cassava. It is well written and clear.

Maybe for such a short communication there are too many references.

I need the authors to provide a better explanation of their approach.

It seems to me they specifically searched for lactobacilli - however, the literature cited in the method description for PCR concerns yeasts. I cannot understand if they also found other microorganisms and yeasts, if they did not find any or if they focused the analysis only on LAB.

The conclusions are not satisfactory in providing answers to the possible health risks for consumers linked to amines.

Author Response

Response to Reviewers’ comments

  1. Ref. No.: microoganisms-1618451

Title: “Lactic acid bacteria and bioactive amines identified during manipueira fermentation aiming tucupi production”

Dear Editor,

Attached, please find the revised manuscript with the requested clarifications and changes. We thank the editor and reviewers for taking the time and for the careful reading and evaluation of this manuscript. We appreciated the suggestions and we are certain that the manuscript has been enhanced with the changes.

Our responses to the recommendations are listed follows.

Best regards.

Reviewer 3

  1. Comment: The paper addresses isolation of bacteria and amines from a traditional procedure on cassava. It is well written and clear.

Our reply: We appreciate the positive feedback from the reviewer. Thank you for your comment and time to evaluate our manuscript.

  1. Comment: Maybe for such a short communication there are too many references.

Our reply: This is a good point. Thank you. We have reduced the number of references from 47 to 29.

  1. Comment: I need the authors to provide a better explanation of their approach.

Our reply: Thank you. A better explanation was made and can be found in page 6 (lines 49-54) and page 7 (lines 1-7).

  1. Comment: It seems to me they specifically searched for lactobacilli - however, the literature cited in the method description for PCR concerns yeasts. I cannot understand if they also found other microorganisms and yeasts, if they did not find any or if they focused the analysis only on LAB.

Our reply: This is a good point that we reviewed. The PCR conditions used in this study were also successfully used for bacteria (Zimmermann et al.). This info was improved in the manuscript (page 3, lines 31-34).

Ref.:

  • Zimmermann, J.; Gonzalez, J.M.; Saiz-Jimenez, C.; Ludwig, W. Detection and phylogenetic relationships of highly diverse uncultured acidobacterial communities in Altamira Cave using 23S rRNA sequence analyses. J. 2005, 22, 379–388, doi.org/10.1080/01490450500248986.

  1. Comment: The conclusions are not satisfactory in providing answers to the possible health risks for consumers linked to amines.

Our reply: The conclusions were improved as requested (page 7, lines 12-23).

Round 2

Reviewer 2 Report

the modifications introduced have significantly improved the work. But, in the next study the authors should verify the level of recovery of the amines in the sample